# Are Consumers Equally Willing to Pay More for Brands That Aim for Sustainability, Positive Societal Contribution, and Inclusivity as for Brands That Are Perceived as Exclusive? Generational, Gender, and Country Differences

Frank Goedertier [1], Bert Weijters [2,*] and Joeri Van den Bergh [3]

1 Area Marketing & Sales, Vlerick Business School, 3000 Leuven, Belgium; frank.goedertier@vlerick.com
2 Department Work, Organization and Society and BE4Life, Faculty of Psychology & Educational Sciences, Ghent University, 9000 Gent, Belgium
3 Human8, 9032 Ghent, Belgium; joeri@wearehuman8.com
* Correspondence: bert.weijters@ugent.be

**Abstract:** This study explores consumer preferences for brands that emphasize sustainability and inclusivity, and for brands perceived as exclusive and trendy. Consumer data obtained via a large-scale survey involving 24,798 participants across 20 countries and one special administrative region (SAR) are used to understand how willingness to pay (WTP) for these brand types varies globally, accounting for demographic factors like generation, gender, and country. A substantial body of literature highlights growing consumer interest in brands that stand for sustainability and inclusivity, challenging traditional notions that luxury and exclusivity primarily drive brand value. Despite persistent skepticism among some business executives about consumers' actual versus claimed willingness to spend more for sustainable and inclusive brands, academics and commercial researchers increasingly signal a shift in purchasing behavior that is influenced by socio-ecological factors. This research aims to provide empirical data on consumer WTP across different demographics and countries/regions, thereby contributing to academic discussions and offering insights for managerial decision making. The study frames its investigation around four research questions, to explore how consumers' WTP for exclusive and inclusive brands varies across generations, genders, and countries/regions. It employs a robust methodological approach, using confirmatory factor analysis (CFA) and structural equation modeling (SEM) to analyze the data. This ensures that the constructs of brand inclusiveness and exclusivity are comparable across diverse cultural contexts. Significant gender, generational, and country/region differences are observed. When comparing generations, the findings indicate that GenZ consumers have a higher WTP for sustainable/inclusive brands (compared to older, GenX, and Baby Boomer generations). Similar patterns are found when considering WTP for exclusive, on-trend brands. In terms of gender, women are observed to have a higher WTP for sustainable/inclusive brands, but a lower WTP for exclusive, on-trend brands compared to men. Finally, compared to consumers originating from certain European countries, we find that consumers living in certain Asian countries/regions have a significantly higher WTP for inclusive and sustainable brands, as well as for exclusive/on-trend brands. The study underscores the complexities of consumer behavior in the global market, highlighting the coexistence of traditional preferences for exclusive, trendy brands and preferences for brands that embrace sustainability and inclusivity.

**Keywords:** sustainable brands; inclusive brands; exclusive brands; willingness to pay (WTP); consumer preferences

## 1. Introduction

Increasingly, it has been shown that consumers value brands that commit to engaging in sustainable business, to impact society in a positive way, and that stand for "inclusivity"

(i.e., respect all individuals across gender, race, sexuality, age, disabilities, etc., and treat all fairly). To illustrate, business research executed in 2021 indicated that 50% of consumers declared sustainability to be a top-five value driver during their purchase process [1]. Yet, in an article by Forbes, it is suggested that certain business studies indicate that over 60% of business executives are skeptical about this [2]. The author concludes that there is a misalignment between consumers and business executives when it comes to perceptions of willingness to pay for inclusive, sustainable brands. Traditionally, many companies have invested into creating brands that are perceived as "exclusive" (i.e., trendy, luxurious, high-status, etc.). A core motivation for this is that there is the conviction (and empirical evidence) that such brands trigger a high willingness to pay a premium [3,4].

It is a widespread belief that purchasing behavior is increasingly not just influenced by functional and emotional criteria but also by socio-ecological factors [5]. However, companies may be more inclined to act on this when they are presented with more evidence that "exclusive" brand positioning as well as "inclusive" brand positioning can lead to an increased willingness to pay. Companies may particularly benefit from specific information regarding which consumer groups are prepared to pay premiums for inclusive, sustainable brands. By offering data-supported insights originating from a large multi-country survey, we aim to provide such information, and contribute at a managerial level. This might also help motivate more inclusive/sustainable brand positionings, thus promoting more effective efforts toward sustainability. To summarize, the research need that this study addresses is the necessity of advancing additional evidence of consumer willingness to pay for inclusive, sustainable offerings of specific customer groups (to increasingly trigger brand positioning on inclusivity and sustainability by businesses). Specifically, we aim to contribute by examining whether "inclusive" brand positioning can trigger similar increased willingness to pay as "exclusive" brand positioning. And if so, for which customer groups especially.

The observations reported are based on data obtained via a survey involving close to 25 k respondents (N = 24,798). This survey was administered in multiple countries and regions across the world, covering 20 countries and one special administrative region (SAR), namely Australia, Argentina, Belgium, Brazil, China (including Hong Kong SAR), Colombia, Germany, France, the United Kingdom, Indonesia, South Korea, Mexico, The Netherlands, the Philippines, Sweden, Singapore, Thailand, Taiwan, the United States, and South Africa.

At an academic level, we aim to contribute to the growing literature stream focusing on willingness to pay for sustainability/inclusivity-focused offerings (e.g., [6–10]). We particularly want to add to specific research sub-streams within this domain: (1) research that focuses on GenZ customers and generational differences (e.g., [11–15]), (2) research that considers gender differences (e.g., [16–18]), and (3) research that studies country differences (e.g., [19,20]). Academic researchers within these research streams have expressed that there is a lack of/opportunity for research on corporate social responsibility (CSR) aspects that considers young(er) persons (e.g., [21–24]). Several researchers have observed a higher inclination among female (vs. male) consumers to engage in sustainable consumption, but some others indicate that opposite results or non-effects have also been observed (e.g., [25]) and call for more research on the gender variable in terms of its relationship with sustainable purchasing behavior. Finally, multiple studies focusing on sustainable purchasing behavior mention a single country/culture focus as a limitation and call for studies that include multiple regions/countries (e.g., [11,13,18,26]). With the research we present, we aim to contribute to addressing these concerns.

To summarize, we add to this literature stream by studying demographic differences in willingness to pay for brands that are offering sustainable propositions, are impacting society in a positive way, and act in an inclusive way (i.e., respecting all individuals across gender, race, sexuality, age, disabilities, etc., and treating all fairly). Specifically, we research to what extent consumers of different countries, different generations, and different genders indicate to be inclined to spend more for such brands. In the remainder

of this paper, we refer to these brands as "inclusive/sustainable brands". We also examine and report the extent to which these consumers are willing to pay (more) for "exclusive/on-trend brands". We use "exclusive/on-trend brands" as an umbrella notion to refer to those brands that offer exclusiveness, are on-trend/trendsetting, and collaborate with celebrities/influencers. We want to stress that "brand inclusivity" and "brand exclusivity" are not conceptualized in this paper as extremes of a same continuum, as can be derived from the definition/description in the first part of this paragraph. We rather perceive these notions as two separate dimensions where a brand can position itself on/is perceived along.

## 2. Embedding the Literature on Willingness to Pay (WTP) for Exclusive Offerings

Several companies have traditionally invested into creating brands that are perceived as exclusive and on-trend/trendsetting. One of the main reasons is that such brands trigger a high willingness to pay a premium [3,4]. This phenomenon has been explained by referring to the social value that such brands offer [27]. Specifically, the consumption of such offerings may signal "having status and wealth" to others [28]. Research has discussed the practice of "lifestyle advertising" (i.e., advertisements that emphasize intangible or abstract features (e.g., lifestyle, personality, taste, identity)) [29] as a road to creating exclusive, luxurious associations around a brand and thus enhancing consumers' willingness to pay a premium [30]. Some researchers have indeed observed an impact of lifestyle advertising on the creation of "exclusivity" associations and willingness to pay more (e.g., [31], who focus on luxury fashion brands in China). Others have also recommended to proceed carefully, considering consumer purchase stages and consumer psychology insights, when aiming for beneficial effects of lifestyle advertising on willingness to pay more (see [29,32] for an interesting construal level theory perspective and consideration of the consumer purchase journey). To summarize, via lifestyle advertising or other means, multiple companies have traditionally attempted to create an "exclusive" image around their brands as they are convinced that this will translate to consumer willingness to pay more. In our study, we examine whether (and if so, which) consumers are equally willing to pay for inclusive/sustainable brands as for exclusive/on-trend brands. Demonstrating similar levels of WTP for sustainable/inclusive brands may offer companies an additional incentive to launch such offerings.

## 3. Embedding the Literature on Willingness to Pay (WTP) for Inclusive, Sustainable Offerings

High prices have been identified in the past as a key obstacle hindering sustainable consumption [33–35]. Sustainable, environmentally friendly products tend to have a higher price tag (vs. comparable "traditional" products) due to a typically higher cost of production [36]. Due to the "high price" element and the increase in sustainable offerings on the market, the challenge of managing WTP is considered a crucial element in the marketing of such offerings [37]. Given the importance of the WTP factor, a specific stream of the literature within the domain of marketing sustainable, inclusive offerings has focused on this. We highlight some previous research findings within this stream and use these to advance a series of research questions that we examine in our research. Specifically, given the focus of our research project, we discuss research that focuses on WTP for sustainable, inclusive offerings of (1) GenZ and (2) women vs. men. We also highlight past research that (3) advances country-specific observations.

### 3.1. The WTP for Inclusive, Sustainable Offerings of NextGen Consumers

Business research by McKinsey indicates that Generation Z (GenZ) is presumed to become one of the largest cohorts of consumers in the world and may represent more than 25% of the global workforce in 2025 [38]. Most definitions consider persons born between the mid-to-later-1990s and 2010 (sometimes expanded to the early 2010s) as belonging to GenZ [11,22,38,39]. In the current study, we label generations as "Baby Boomer" if born

between 1946 and 1964, "GenX" if born between 1965 and 1979, "GenY" if born between 1980 and 1996, and "GenZ" if born between 1997 and 2011.

Young (GenZ) consumers, whose conscious awareness and needs intensified due to the pandemic context and social movements, are more concerned about the impact of their decisions on the ecosystem [40,41]. Both academic sources and leading business magazines such as Forbes put forth that GenZ expects brand action on sustainability (e.g., [42,43]). Business research executed at the end of 2018 already indicated that GenZ (compared to consumers of other generations in the sample) is three times more likely to express "...that the purpose of business is to serve communities and society rather than to simply make good products and services" [44]. In contrast to GenY, GenZ was confronted with environmental and social concerns at an early age and has been raised in a context of fast change [39,45]. In addition, it has been observed that the effect of the "...perceived popularity among peers of sustainable and/or inclusive marketing practices" on attitudes towards such practices of others is stronger within the GenZ cohort (compared to GenY) [46]. These are some generation-specific elements that suggest that addressing social unrest and engaging in global climate change initiatives are likely to be more salient for GenZ (vs. other generations) [13,44]. Other observations and claims in previous academic and business studies in the literature support this view. For example, referring to research by Forbes, Petro [47] indicates that 62% of GenZ individuals prefer to buy from sustainable brands. In the same vein, Song et al. [43] indicate that GenZ is more socially and economically active than earlier generations and appreciates sustainability. Damico et al. [48] surveyed GenZ "Zoomer" consumers in Argentina and observed that they express a high concern for the health of the planet and for unsustainable production methods. Given the documented social and environmental responsibility concerns of GenZ in the previous literature, an effect on WTP can be expected. Although GenZ is often referred to as being green-friendly and willing to spend on green offerings (e.g., [12,49]), to our knowledge, the extant academic research that actively researches this topic (i.e., the WTP for inclusive, sustainable offerings of GenZ) is rather scarce. We discuss two relatively recent publications.

A first study by Narayanan [13] used real Fast-Moving Consumer Good (FMCG) brands with actual CSR data as the stimuli. GenZ students belonging to a sample, recruited from three universities in western India (N = 414), were requested to indicate their WTP for these brands with clear environmental or social CSR perceptions. The results were compared with a control group. The research shows that GenZ values CSR, leading to higher WTP for both lesser-known and well-known brands. There was no significant difference in WTP between brands with environmental perceptions vs. those with social CSR perceptions, nuancing previous beliefs about GenZ's preferences. Economic CSR perceptions, however, did not trigger a higher WTP.

A study by Gomes, Lopes, and Nogueira [11] examined to what extent certain determinants impact the WTP of GenZ for sustainable/inclusive products. Analysis of the survey responses of a sample of N = 708 Portuguese GenZ participants (derived from a larger initial sample) revealed that "environmental concerns" primarily influenced GenZ's WTP for green products, followed by "perceived (product) quality" and "future expectations" (i.e., the impression that living in a sustainable/inclusive manner is the way of the future). Contrary to expectations, "perceived (product) benefits" (i.e., positive perceptions of health, taste, and flavor) had a negative impact on WTP. The authors also report a slight increase in WTP with age within the GenZ sample studied.

Building on the insights of the above studies and on related observations reported in the academic literature on sustainable/inclusive consumption, we formulate the following research question (RQ):

*RQ1—Compared to other generations, do younger age cohorts, and especially GenZ consumers, express more willingness to pay for brands that are offering sustainable propositions, are impacting society in a positive way, and are standing for inclusivity (i.e., respecting all individuals across gender, race, sexuality, age, disabilities, etc., and treating all fairly)?*

Our study addresses some limitations of the studies discussed in the paragraphs above. While Narayanan's [13] research was confined to a specific demographic in India and Gomes, Lopes, and Nogueira [11] focused solely on Portuguese GenZ individuals, our study provides a broader perspective by incorporating diverse geographical areas and generational cohorts. This comprehensive approach allows us to examine cross-cultural and age-related (between-generation) variations in consumer behavior, offering a more nuanced understanding of global trends in the WTP for sustainable and inclusive offerings.

*3.2. Gender Differences in the WTP for Inclusive, Sustainable Offerings*

From a business perspective, it is of interest for customer profiling reasons to gain insights into possible gender differences related to purchasing behavior-related variables for inclusive/sustainable offerings. This is reflected in business publications that focus on socio-demographics when discussing trends related to inclusive/sustainable consumer behavior. For example, a study by McKinsey [50] highlights survey data (obtained from a cross-generational sample (N = 4912) with respondents from Germany, Switzerland, and Austria) that suggest that women (vs. men) are in general slightly more willing to pay more for sustainable/inclusive offerings (52% vs. 49% indicate to be willing to pay a premium, but results are mixed when considering different product categories).

Academic research focusing on purchasing-related variables in the context of the consumption of inclusive/sustainable offerings also often explicitly considers the gender element. For example, research by Jekanowski et al. [51], consisting of a survey in the USA, Indiana (N = 320), observes that women (vs. men) are more inclined to buy locally produced agricultural products. A study conducted in the UK [52] finds a higher purchase likeliness of women (vs. men) in the context of buying organic food. Fisher et al. [53] indicate that gender significantly impacts sustainable-focused purchasing behavior: women are observed to be more inclined to buy environmentally friendly products. While several previous academic research findings seem to point to a higher purchase intention of inclusive/sustainable offerings for women (vs. men), as illustrated by the examples mentioned above, other research observes opposite or null effects. To illustrate, Mair [54] examines carbon offsets in an air travel business setting and finds that the purchasing behavior of males (vs. females) is more likely to be influenced by such initiatives. Choi and Ritchie [55] observe no effect of gender (and other elements) when attempting to explain differences in purchasing-related variables for carbon offsets by looking at customer profiles.

As our key focus in this study is on a specific purchasing-related variable (WTP), we also highlight some academic studies that considered this specific angle and included observations related to gender (the focus of this section).

Using experimental auctions with Texas (USA) student subjects (N = 69), Hustvedt and Bernard [16] found that women (vs. men) were more willing to pay for organic apparel products (specifically "socks" made from organic fiber). Their findings suggest gendered trends in the valuation of sustainable clothing. Hinnen et al. [25] did not find significant gender differences in the WTP for supplementary green air travel services (e.g., carbon offsets, organic on-board food) that are sold on top of the travelling ticket. Their study, in which Swiss airline passengers (N = 811) were surveyed, indicated a uniformity in eco-consciousness across genders in this context. Khan et al. [17] found a slightly higher willingness to pay for socially responsible food among men (vs. women) in a sample of consumers from India (N = 398). Their results hint at a gender-related variation in sustainable food purchasing behavior within this market. The observations, identifying men as most willing to spend, contrast with most studies that observe women to have a higher WTP for inclusive/sustainable offerings. Dangelico et al. [26] observed that Italian women are more likely than men to pay a premium for "Made in Italy" and "organic" products after the COVID-19 pandemic. Their study consisted of a survey with N = 1535 participants. Their findings indicate gender differences in sustainable-spending-readiness that are in line with those typically observed in other regions/for other product categories (i.e., with women behaving most sustainably). Shahsavar et al. [18] report that

Czech women are willing to pay more for eco-friendly furniture compared to men. They surveyed clients of a high-quality furniture store (N = 195) and found that female customers are willing to pay on average USD 119.5 (CZK 3000) more than men for an eco-friendly sofa that would 'normally' cost USD 398.35 (CZK 10,000). These results indicate a clear gender-based preference for sustainability in the furniture market.

To summarize, gender has been identified in most of the previous research as a significant demographic predictor for environmentally friendly buying behavior. It also has been observed that mostly women (vs. men) tend to behave more sustainably (see examples above), but this finding is not consistent in all studies. Starting from these insights, we formulate the following research question:

*RQ2—Across generations and countries, do female consumers express more willingness to pay for brands that are offering sustainable propositions, are impacting society in a positive way, and are standing for inclusivity (i.e., respecting all individuals across gender, race, sexuality, age, disabilities, etc., and treating all fairly)?*

As already mentioned, our study aims to address some limitations that are highlighted by previous research. In this section that reviews the literature on gender differences in the WTP for inclusive, sustainable offerings, a specific country focus was a recurring limitation that was frequently mentioned (as was also the case in the research stream focusing on NextGen and WTP for such offerings—see above). To address this element, we present a study in which 20 countries and one special administrative region (SAR) are involved, as already mentioned before. Given this multi-country dimension of our study, we also include a final theoretical section in which we highlight previous research that also considered multiple countries and focused on geographical and cultural differences with respect to the focal subject of this study (i.e., WTP for inclusive, sustainable offerings).

### 3.3. Country/Culture Effects in the Context of the WTP for Inclusive, Sustainable Offerings

Business research by GFK from mid-2023 [56] compares the WTP for sustainable offerings of consumers originating from the UK, Germany, Italy, and France. The data indicate that the WTP for sustainable offerings is highest in Germany and the most stable considering the previous measurements. Over time, the WTP for sustainable offerings has increased the most in the UK (compared to the other countries studied). France also shows a positive trend. Italy, on the contrary, is reported to have a decline in WTP for sustainable offerings. The persistence of high cost of living concerns, especially for everyday purchases, is advanced by GFK as a major reason. Some other similar major business research initiatives that focus on sustainable consumption also offer a multi-country perspective.

To illustrate, the Global Sustainability Study 2021 survey of Simon-Kucher and partners [1] included data of over 10.000 consumers originating from/active in 17 countries: the USA, Germany, Denmark, Sweden, Brazil, China, Japan, Spain, Switzerland, the UK, Australia, Austria, France, The Netherlands, Norway, Italy, and the UAE. An article about the most recent version of this Global Sustainability Study [57] highlights that 37% of global respondents are prepared to pay more for sustainable offerings. In terms of country and regional differences, the study indicates that consumers in the Asia–Pacific (APAC) region (vs. Europe and North America) are considerably more willing to pay a premium for sustainable offerings. For example, for consumer goods and services, APAC respondents are willing to spend 55% more on sustainable offerings (for North American respondents, this is 36%, and for European respondents, 32%). The study also points to the impact of a decreased share-of-wallet to spend on sustainable alternatives due to inflation. This phenomenon is universally acknowledged by consumers as 90% of APAC and North American respondents and 85% of European respondents indicate that price increases and inflation impact their decision making regarding sustainable offerings.

Although many academic studies have focused on one region/country, some academic research has also considered multiple countries and studied regional differences related to sustainability behaviors. We include an overview of past research, originating from a recent

paper by Shehawy et al. [20]. The authors provide a list of studies that demonstrated country differences regarding pro-environmental attitudes. Hudson and Ritchie [58] focused on the UK, Canada, and the USA and observed an effect of national culture on skiers' attitudes towards the environmental impact of skiing. Bohdanowicz [59] focused on Sweden and Poland and found an effect of country origin on the pro-environmental attitudes of hotel operators. Kang and Moscardo [60] studied South Korea, the UK, and Australia and reported an effect of national culture on attitudes towards responsible tourist behavior. Landauer et al. [61] focused on Australia and Finland and found an impact of country on skiers' preferences of climate change adaptation strategies in a skiing destination. Xu and Fox [62] studied the UK and China and observed differences between those countries in the understanding of sustainable management practices in national parks. Packer et al. [63] focused on China and Australia and found (among others) country differences in tourist attitudes towards environmental issues. Kim and Filimonau [64] studied South Korea and China and found that language shapes the attitude of tourists towards environmental impacts. Filimonau et al. [64] focused on Poland and found that the national cultural dimensions/values of "individualism", "long-term orientation", and "harmony" impact the pro-environmental attitudes of tourists. Finally, He and Filimonau [65] studied the UK and China and highlighted causal relationships between tourists' cultural backgrounds, their environmental knowledge, pro-environmental attitudes, and pro-environmental behavioral intentions.

Given our key focus on WTP for inclusive/sustainable offerings, we also highlight some academic studies that examined the impact of country/cultural differences on this outcome variable.

Gregory-Smith, Manika, and Demeril [19] analyzed a 2012 EU survey (N = 21,514) that examined the WTP for the sustainable offerings of the inhabitants of the 28 countries that made up the European Union (EU) at that time. They find (among others) that Southern and Eastern EU countries (S-E) have higher levels of WTP for sustainable offerings, compared to Northern and Western EU countries (N-W). The authors argue that at first, higher levels of WTP and stronger positive attitudes towards sustainable offerings in S-E EU countries may come across as strange, given the lower levels of GDP in these countries. However, they draw on the (eco)innovation diffusion literature [66] to argue that in markets where sustainable offerings are not widely available (here: for S-E EU countries), companies that offer such options may command price premiums due to associations of novelty and higher status (in addition to the "green" association) [67,68]. On the other hand, when sustainable offerings become widespread (here: for N-W EU countries), the opportunity to claim price premiums declines.

Another recent research project that adopts a multi-country angle in the context of WTP for sustainable offerings is authored by Shehawy et al. [20]. They examine the drivers influencing the willingness to pay more for sustainable hotels across seven countries (i.e., the UK, the USA, France, Turkey, China, South Korea, and Egypt). We focus on the results related to country differences. Traditional survey research (study 1; N = 5270 respondents) reveals significant differences in the WTP for sustainable offerings in the United States–China, the United Sates–Turkey, the United Kingdom–China, and Korea–Turkey comparisons. A follow-up telephone survey (study 2), for which 3650 respondents of study 1 were successfully re-contacted (5 months after study 1), examined how many times the respondents had actually stayed in a sustainable hotel in the preceding five months. A clear correlation was observed between the willingness (of hotel guests) to pay more for sustainable hotels (measured in study 1) and their actual behaviors (measured in study 2). In the conclusion section, the authors highlight the distinctive cultural settings of emerging economies such as China, Egypt, and Turkey when discussing the observed differences in WTP for sustainability.

Considering these previous research findings, we formulate the following research question:
*RQ3—Across generations and genders, are there country differences in willingness to pay for brands that are offering sustainable propositions, are impacting society in a positive way, and are standing*

*for inclusivity (i.e., respecting all individuals across gender, race, sexuality, age, disabilities, etc., and treating all fairly), especially between established and emerging economies (RQ3a)?*

## 4. Methods

### 4.1. Sample

For the current study, we analyzed secondary data from a large-scale cross-national online survey conducted among the online panel of a globally operating market research and consulting agency using quota sampling for the country, gender, and age cohorts (oversampling GenZ as this generation was a core focus of the survey study). The company selected the countries based on commercial relevance. Table 1 provides a demographic breakdown of the sample (N = 24,798 across 20 countries and one special administrative region (SAR)) by country, age cohort, and gender.

**Table 1.** Sample composition.

|  | Age Cohort | Male | Female | Total |
|---|---|---|---|---|
| Australia | Baby Boomers | 192 | 236 | 428 |
|  | GenX | 168 | 220 | 388 |
|  | GenY | 140 | 262 | 402 |
|  | GenZ | 426 | 587 | 1013 |
|  | Total | 926 | 1305 | 2231 |
| Argentina | Baby Boomers | 98 | 102 | 200 |
|  | GenX | 101 | 100 | 201 |
|  | GenY | 98 | 101 | 199 |
|  | GenZ | 200 | 200 | 400 |
|  | Total | 497 | 503 | 1000 |
| Belgium | Baby Boomers | 130 | 86 | 216 |
|  | GenX | 90 | 106 | 196 |
|  | GenY | 90 | 109 | 199 |
|  | GenZ | 313 | 269 | 582 |
|  | Total | 623 | 570 | 1193 |
| Brazil | Baby Boomers | 101 | 100 | 201 |
|  | GenX | 94 | 106 | 200 |
|  | GenY | 98 | 101 | 199 |
|  | GenZ | 184 | 216 | 400 |
|  | Total | 477 | 523 | 1000 |
| China | Baby Boomers | 181 | 129 | 310 |
|  | GenX | 250 | 142 | 392 |
|  | GenY | 443 | 362 | 805 |
|  | GenZ | 517 | 499 | 1016 |
|  | Total | 1391 | 1132 | 2523 |
| Colombia | Baby Boomers | 107 | 93 | 200 |
|  | GenX | 100 | 100 | 200 |
|  | GenY | 99 | 101 | 200 |
|  | GenZ | 198 | 202 | 400 |
|  | Total | 504 | 496 | 1000 |
| Germany | Baby Boomers | 120 | 95 | 215 |
|  | GenX | 87 | 110 | 197 |
|  | GenY | 93 | 114 | 207 |
|  | GenZ | 263 | 254 | 517 |
|  | Total | 563 | 573 | 1136 |

**Table 1.** *Cont.*

|  | Age Cohort | Male | Female | Total |
|---|---|---|---|---|
| France | Baby Boomers | 118 | 92 | 210 |
|  | GenX | 69 | 137 | 206 |
|  | GenY | 72 | 122 | 194 |
|  | GenZ | 277 | 267 | 544 |
|  | Total | 536 | 618 | 1154 |
| UK | Baby Boomers | 111 | 105 | 216 |
|  | GenX | 69 | 131 | 200 |
|  | GenY | 74 | 120 | 194 |
|  | GenZ | 272 | 261 | 533 |
|  | Total | 526 | 617 | 1143 |
| Hong Kong SAR | Baby Boomers | 67 | 59 | 126 |
|  | GenX | 146 | 140 | 286 |
|  | GenY | 161 | 223 | 384 |
|  | GenZ | 84 | 120 | 204 |
|  | Total | 458 | 542 | 1000 |
| Indonesia | Baby Boomers | 88 | 59 | 147 |
|  | GenX | 99 | 122 | 221 |
|  | GenY | 117 | 118 | 235 |
|  | GenZ | 202 | 196 | 398 |
|  | Total | 506 | 495 | 1001 |
| South Korea | Baby Boomers | 89 | 116 | 205 |
|  | GenX | 115 | 100 | 215 |
|  | GenY | 109 | 102 | 211 |
|  | GenZ | 179 | 190 | 369 |
|  | Total | 492 | 508 | 1000 |
| Mexico | Baby Boomers | 108 | 74 | 182 |
|  | GenX | 111 | 105 | 216 |
|  | GenY | 85 | 117 | 202 |
|  | GenZ | 198 | 202 | 400 |
|  | Total | 502 | 498 | 1000 |
| The Netherlands | Baby Boomers | 127 | 80 | 207 |
|  | GenX | 82 | 126 | 208 |
|  | GenY | 97 | 110 | 207 |
|  | GenZ | 245 | 285 | 530 |
|  | Total | 551 | 601 | 1152 |
| Philippines | Baby Boomers | 81 | 109 | 190 |
|  | GenX | 98 | 105 | 203 |
|  | GenY | 102 | 107 | 209 |
|  | GenZ | 202 | 196 | 398 |
|  | Total | 483 | 517 | 1000 |
| Sweden | Baby Boomers | 107 | 109 | 216 |
|  | GenX | 91 | 103 | 194 |
|  | GenY | 88 | 117 | 205 |
|  | GenZ | 302 | 267 | 569 |
|  | Total | 588 | 596 | 1184 |
| Singapore | Baby Boomers | 98 | 93 | 191 |
|  | GenX | 99 | 101 | 200 |
|  | GenY | 105 | 104 | 209 |
|  | GenZ | 196 | 204 | 400 |
|  | Total | 498 | 502 | 1000 |

**Table 1.** *Cont.*

|  | Age Cohort | Male | Female | Total |
|---|---|---|---|---|
| Thailand | Baby Boomers | 92 | 108 | 200 |
|  | GenX | 99 | 101 | 200 |
|  | GenY | 99 | 109 | 208 |
|  | GenZ | 195 | 197 | 392 |
|  | Total | 485 | 515 | 1000 |
| Taiwan | Baby Boomers | 86 | 59 | 145 |
|  | GenX | 112 | 122 | 234 |
|  | GenY | 115 | 110 | 225 |
|  | GenZ | 207 | 196 | 403 |
|  | Total | 520 | 487 | 1007 |
| USA | Baby Boomers | 82 | 124 | 206 |
|  | GenX | 75 | 125 | 200 |
|  | GenY | 92 | 106 | 198 |
|  | GenZ | 204 | 264 | 468 |
|  | Total | 453 | 619 | 1072 |
| South Africa | Baby Boomers | 82 | 117 | 199 |
|  | GenX | 95 | 107 | 202 |
|  | GenY | 106 | 95 | 201 |
|  | GenZ | 200 | 200 | 400 |
|  | Total | 483 | 519 | 1002 |

*4.2. Procedure and Measures*

The respondents filled out an online survey containing a variety of questions, including the demographics gender ("Which gender do you identify with most? Male, female, non-binary, other, prefer not to say") and year of birth. The survey then proceeded with a series of questions relating to general consumption behaviors and preferences. These questions were generated starting from the insights of previous qualitative and quantitative studies (i.e., a qualitative study based on 26 expert interviews with senior marketing executives of youth-targeting brands [69], a quantitative study involving 10,000 respondents originating from 8 European countries [70], and an interpandemic qualitative study involving 16- to 19-year-olds (N = 200) originating from 8 European countries during a 3 week online community) [71]. This study focuses on a question in which respondents rated their willingness to pay for brands ("To what extent do you agree with the following statements? I am willing to pay more for brands that…"), followed by four items that capture WTP for inclusive, sustainable brands ("… run their business in a sustainable way", "… try to have a positive impact on society", "… use inclusive practices, respecting all individuals across gender, race, sexuality, age, disabilities, etc.", "… treat employees and suppliers fairly") and three items that capture WTP for exclusive, on-trend brands ("… collaborate with celebrities/influencers, "… offer exclusiveness", "… are on trend/setting the trend"). Each criterion was rated on a five-point rating scale ranging from 1 = I definitely do not agree to 5 = I definitely agree.

**5. Results**

As a first analytical step, we specify a multi-group confirmatory factor analysis (CFA), using country as the grouping variable. There are two factors, which we refer to in this section as "brand inclusiveness" and "brand exclusivity" (see Table 2). To ensure that the factors can be interpreted equivalently across countries, we first test for measurement invariance.

**Table 2.** Items by factor.

| Factor Name | CR; AVE | Item | SL |
|---|---|---|---|
| WTP inclusive, sustainable brands | 0.84; 0.56 | . . . run their business in a sustainable way | 0.76 |
| | | . . . try to have a positive impact on society | 0.78 |
| | | . . . use inclusive practices, respecting all individuals across gender, race, sexuality, age, disabilities, etc. | 0.72 |
| | | . . . treat employees and suppliers fairly | 0.74 |
| WTP exclusive, on-trend brands | 0.79; 0.55 | . . . collaborate with celebrities/influencers | 0.70 |
| | | . . . offer exclusiveness | 0.74 |
| | | . . . are on trend/setting the trend | 0.79 |

Note: CR = composite reliability; AVE = average variance extracted; SL = standardized factor loading in the USA sample in the partial scalar invariance model.

Measurement invariance testing is integral to validating the comparability of factor structures across groups. This involves a hierarchical testing procedure, starting from configural invariance (same pattern of fixed and free parameters) to the more stringent levels of metric invariance (equivalence of factor loadings) and scalar invariance (equivalence of intercepts). The evaluation of invariance was informed by several fit indices, including the Chi-square ($\chi^2$) test, Comparative Fit Index (CFI), Tucker–Lewis Index (TLI), Standardized Root Mean Square Residual (SRMR), and Root Mean Square Error of Approximation (RMSEA). However, given the sensitivity of these indices to sample size and model complexity, the Bayesian Information Criterion (BIC) was employed as a primary criterion for judgment, in line with the methodological recommendations [72,73]. Lower BIC values across increasingly restrictive invariance models indicate a better balance between model fit and complexity, thereby affirming measurement invariance without unduly penalizing model parsimony. This nuanced approach facilitates a robust assessment of the equivalence of constructs across diverse cultural contexts, ensuring the reliability and validity of the conclusions drawn about consumer preferences for brand inclusiveness and brand exclusivity on a global scale. The model fit results are reported in Table 3. Full scalar invariance does not hold, and after the inspection of modification indices, we release the intercept of the item ". . . collaborate with celebrities/influencers" for the China sample (the intercept for this item is significantly higher than in the other samples, $p < 0.001$). After relaxing the equality constraint for this parameter, partial scalar invariance can be accepted (see Table 3). The correlation (all $p < 0.001$) between the WTP for inclusive and exclusive brands is positive in all countries/regions, with r = 0.514 in Argentina, 0.548 in Australia, 0.489 in Belgium, 0.739 in Brazil, 0.775 in China, 0.489 in Colombia, 0.628 in France, 0.557 in Germany, 0.644 in Hong Kong SAR, 0.644 in Indonesia, 0.594 in Mexico, 0.545 in The Netherlands, 0.532 in the Philippines, 0.593 in Singapore, 0.607 in South Africa, 0.728 in South Korea, 0.403 in Sweden, 0.586 in Taiwan, 0.821 in Thailand, 0.600 in the UK, and 0.588 in the US.

As the next step, we develop a multi-group structural equation model, treating "brand inclusiveness" and "brand exclusivity" as outcomes influenced by demographic variables. Specifically, we use dummy variables for gender (1 representing female, 0 representing male) and age groups (dummies for Generation Y, Generation X, and Baby Boomers). To prevent the model from becoming too complex and to ensure it remains straightforward, we check if the influence of these demographic variables on the brand factors is consistent across different countries. This process, known as testing for invariance, helps us ensure that the model is not overly tailored to specific datasets. As indicated in Table 3 and based on the Bayesian Information Criterion (BIC), the relationships between demographics and brand factors do not vary by country, suggesting a stable model across various national contexts.

**Table 3.** Cross-country multi-group model invariance tests.

|  |  | Chi2 | DF | RMSEA | CFI | TLI | SRMR | BIC |
|---|---|---|---|---|---|---|---|---|
| CFA | Configural invariance model | 1333.9 | 273 | 0.057 | 0.983 | 0.973 | 0.029 | 457,716.2 |
|  | Metric invariance model | 1552.2 | 373 | 0.052 | 0.981 | 0.978 | 0.041 | 456,922.6 |
|  | Scalar invariance model | 2823.9 | 473 | 0.065 | 0.962 | 0.965 | 0.056 | 457,182.5 |
|  | Partial scalar invariance model | 2439.5 | 472 | 0.059 | 0.969 | 0.971 | 0.048 | 456,808.2 |
| SEM | Structural model | 3437.2 | 892 | 0.049 | 0.961 | 0.955 | 0.041 | 456,009.7 |
|  | Gender effect invariance | 3660.2 | 932 | 0.050 | 0.958 | 0.954 | 0.043 | 455,827.9 |
|  | Full structural invariance model | 4688.9 | 1052 | 0.054 | 0.944 | 0.946 | 0.059 | 455,642.4 |

Note: The section 'CFA' displays model fit results for the multi-group confirmatory factor analysis. In the partial scalar invariance model, the item intercepts of item "... collaborate with celebrities/influencers" is freely estimated for China (where it is higher). The section 'SEM' displays model fit results for the multi-group structural equation model where the "WTP for inclusive, sustainable brands" and the "WTP for exclusive, on-trend brands" factors are specified as the dependent variables, with gender (a dummy variable for female) and age cohort (dummy variables for GenY, GenX, and Baby Boomers) as the independent variables. In the Gender effect invariance model, the regression weights involving gender are constrained to be equal across groups (countries); in the full structural invariance model, all structural regression weights (i.e., regression weights involving gender or age cohort dummies) are constrained to be equal across groups (countries).

Table 4 and Figure 1 report the effects of age cohort (using GenZ as the reference category with mean zero) and gender (using men as the reference category with mean zero) on WTP for inclusive, sustainable brands and exclusive, on-trend brands. Compared to GenZ, GenX and the Baby Boomer generation show a lower WTP for inclusive, sustainable brands. These same generations also show a lower WTP for exclusive, on-trend brands (especially the Baby Boomer generation). GenY scores similar to GenZ (i.e., slightly but not statistically significantly higher on both factors compared to GenZ). Women show a higher WTP for inclusive, sustainable brands, but a lower WTP for exclusive brands compared to men.

**Table 4.** Effects of age cohort and gender on willingness to pay (WTP) for inclusive, sustainable brands and exclusive, on-trend brands.

| DV | IV | Est. | LL | UL |
|---|---|---|---|---|
| WTP inclusive, sustainable brands | GenY | 0.024 | −0.008 | 0.056 |
|  | GenX | −0.159 | −0.194 | −0.124 |
|  | GenBB | −0.181 | −0.218 | −0.144 |
| WTP exclusive, on-trend brands | GenY | 0.012 | −0.019 | 0.044 |
|  | GenX | −0.292 | −0.329 | −0.256 |
|  | GenBB | −0.516 | −0.56 | −0.472 |
| WTP inclusive, sustainable brands | Female | 0.098 | 0.073 | 0.123 |
| WTP exclusive, on-trend brands | Female | −0.075 | −0.100 | −0.051 |

Note: DV = dependent variable; IV = independent variable; Est. = estimate, LL = lower limit, UL = upper limit of the 95% confidence interval. Estimates are regression coefficients (with the dependent variable standardized) with their 95% confidence interval. GenZ serves as the reference category (with latent mean zero) for age cohort effects, and male serves as the reference category (with latent mean zero) for gender effects.

Table 5 and Figure 2 provide the country latent means (with a 95% confidence interval) of WTP for inclusive, sustainable brands and exclusive, on-trend brands. Countries are alphabetically ordered. As can be seen from these results, Singapore and South Korea are the countries that show the highest WTP for inclusive, sustainable brands, whereas Belgium, Germany, The Netherlands, Sweden, and the UK show the lowest WTP for inclusive, on-trend brands. China, South Africa, and Thailand show the highest WTP for exclusive, on-trend brands, whereas Belgium, Germany, The Netherlands, and Sweden show the lowest WTP for exclusive, on-trend brands.

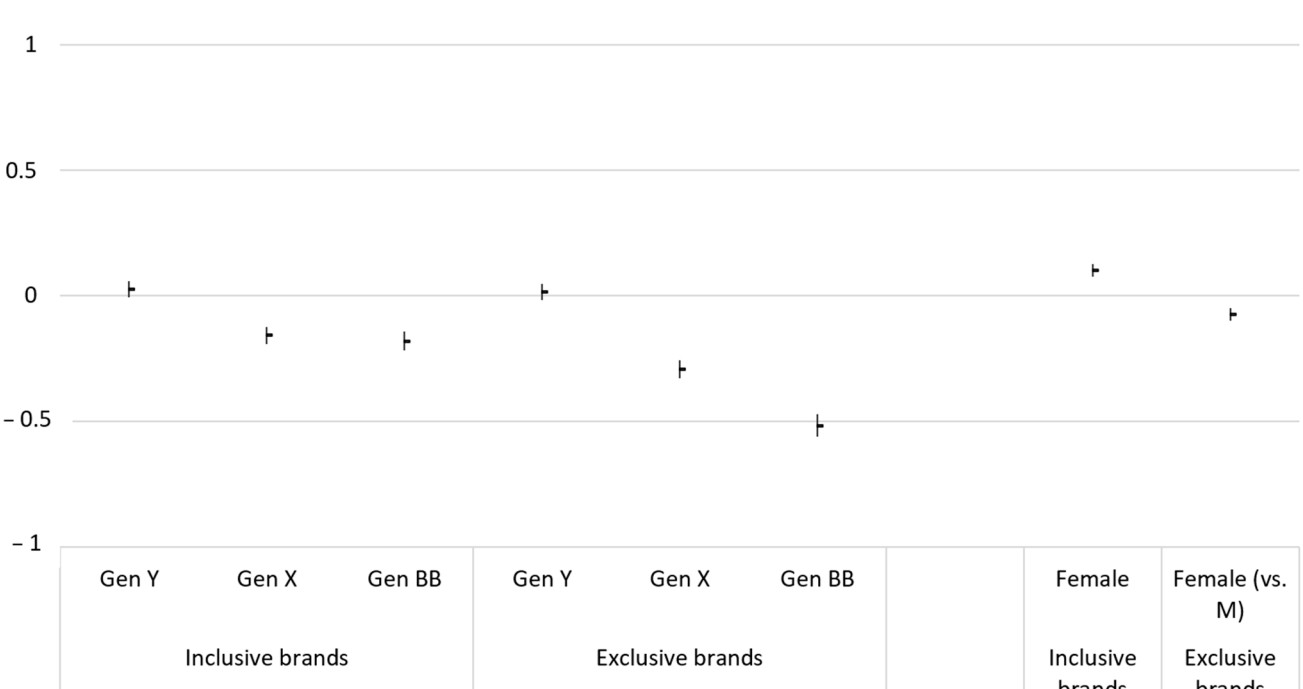

**Figure 1.** Effects of age cohort and gender on willingness to pay (WTP) for inclusive, sustainable brands and exclusive, on-trend brands. Note: Values are regression coefficients (with the dependent variable standardized) with their 95% confidence interval. GenZ serves as the reference category (with latent mean zero) for age cohort effects, and male serves as the reference category (with latent mean zero) for gender effects.

**Table 5.** Country latent means with 95% confidence interval of willingness to pay (WTP) for inclusive, sustainable brands and exclusive, on-trend brands.

|  | Inclusive Brands | | | Exclusive Brands | | |
|---|---|---|---|---|---|---|
|  | Est. | LL | UL | Est. | LL | UL |
| Argentina | 0.088 | 0.008 | 0.168 | −0.028 | −0.112 | 0.057 |
| Australia | −0.158 | −0.225 | −0.091 | −0.152 | −0.224 | −0.081 |
| Belgium | −0.307 | −0.384 | −0.231 | −0.337 | −0.418 | −0.257 |
| Brazil | 0.367 | 0.288 | 0.445 | 0.356 | 0.269 | 0.442 |
| China | 0.201 | 0.142 | 0.261 | 0.614 | 0.546 | 0.681 |
| Colombia | 0.341 | 0.264 | 0.418 | 0.123 | 0.040 | 0.206 |
| France | −0.182 | −0.259 | −0.104 | −0.107 | −0.190 | −0.024 |
| Germany | −0.285 | −0.365 | −0.206 | −0.224 | −0.307 | −0.141 |
| Hong Kong SAR | −0.076 | −0.145 | −0.007 | 0.322 | 0.245 | 0.399 |
| Indonesia | 0.420 | 0.351 | 0.489 | 0.430 | 0.351 | 0.508 |
| Mexico | 0.184 | 0.104 | 0.264 | 0.085 | −0.002 | 0.172 |
| The Netherlands | −0.276 | −0.352 | −0.200 | −0.224 | −0.305 | −0.142 |
| Philippines | 0.601 | 0.531 | 0.671 | 0.441 | 0.360 | 0.522 |
| Singapore | −0.048 | −0.121 | 0.024 | 0.153 | 0.074 | 0.233 |
| South Africa | 0.559 | 0.500 | 0.617 | 0.729 | 0.640 | 0.817 |
| South Korea | −0.070 | −0.142 | 0.002 | 0.320 | 0.245 | 0.395 |
| Sweden | −0.196 | −0.273 | −0.119 | −0.337 | −0.417 | −0.256 |
| Taiwan | 0.176 | 0.106 | 0.247 | 0.341 | 0.264 | 0.418 |
| Thailand | 0.406 | 0.333 | 0.479 | 0.753 | 0.672 | 0.834 |
| UK | −0.226 | −0.303 | −0.148 | −0.114 | −0.195 | −0.033 |

Note: Est. = estimate, LL = lower limit, UL = upper limit of the 95% confidence interval. The USA serves as the reference group by setting its latent mean to zero.

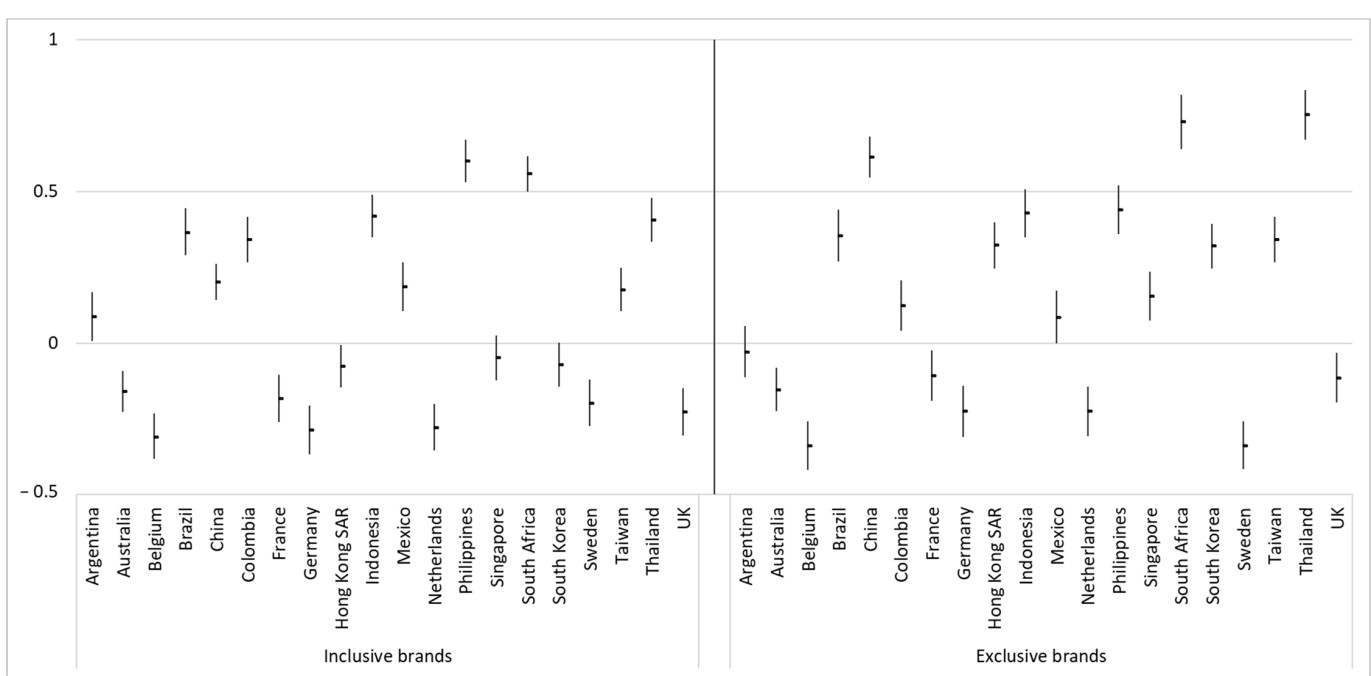

**Figure 2.** Country/region latent means with 95% confidence interval of willingness to pay (WTP) for inclusive, sustainable brands and exclusive, on-trend brands. Note: the USA serves as the reference group by setting its latent mean to zero.

## 6. Discussion

### 6.1. Main Observations

In this study, we analyze secondary data from a large-scale cross-national online survey, which encompassed responses from 24,798 individuals across 20 countries and one special administrative region (SAR), namely Australia, Argentina, Belgium, Brazil, China (including Hong Kong SAR), Colombia, Germany, France, the United Kingdom, Indonesia, South Korea, Mexico, The Netherlands, the Philippines, Sweden, Singapore, Thailand, Taiwan, the United States, and South Africa. A demographic breakdown by country/region, age cohort, and gender allowed for an in-depth analysis of global consumer behavior related to spending-readiness for inclusive, sustainable offerings. Respondents assessed their willingness to pay (WTP) for brands that act sustainably (referred to as "inclusive, sustainable brands") and for brands that are associated with luxury, prestige, and trendiness (referred to as "exclusive, on-trend brands") using a multi-item five-point scale for both factors. The "WTP for inclusive, sustainable brands" factor represents consumer readiness to pay more for brands that prioritize sustainability, social impact, inclusivity across diverse demographics, and fair treatment of employees and suppliers. This reflects a value-driven consumer behavior focusing on ethical and responsible business practices. The "WTP for exclusive, on-trend brands" factor gauges the extent to which consumers are willing to spend more on brands that are associated with celebrities/influencers, offer exclusivity, and are trendsetters, indicating a preference for luxury, status, and fashion-forwardness in their purchasing decisions.

Our study directly addresses the research question concerning whether younger generations, particularly GenZ, show a greater willingness to pay (WTP) for brands that emphasize sustainability, positive societal impact, and inclusivity (RQ1). The data reveal that compared to older generations like GenX and Baby Boomers, GenZ consumers are indeed more inclined to invest in brands that are both inclusive and sustainable, as well as those that are exclusive and trendy. This pattern underscores a significant generational shift in brand preference and consumer values. Specifically, while Baby Boomers show considerably less interest in paying a premium for these brand types, GenZ's equal willing-

ness to support both inclusive, sustainable brands and exclusive, trendy brands suggests a unique consumer profile. This finding suggests that for companies aiming to attract the GenZ market, adopting strategies that promote inclusivity and sustainability could be as effective as those that focus on exclusivity and trendiness. Our results affirm that the traditional focus on exclusivity alone may not suffice for engaging GenZ consumers, who are also drawn to brands that make a positive impact and promote inclusivity across various demographics.

Our research specifically addresses the second research question by examining whether, across different generations and countries, female consumers show a higher willingness to pay (WTP) for brands committed to sustainability, positive societal impact, and inclusivity. Our findings confirm that gender significantly influences consumer preferences, with female consumers demonstrating a greater WTP for inclusive and sustainable brands compared to their male counterparts. This supports the existing research and observed trends indicating that women generally prioritize ethical consumption and sustainability more than men. Conversely, our results also highlight that women are less likely than men to pay a premium for exclusive brands. These insights are crucial for businesses as they suggest that marketing strategies focusing on sustainability and inclusivity may be particularly effective in engaging female consumers globally, reinforcing the need for brands to align their values and operations with the priorities of this demographic.

As to research question 3, our methodological approach involved a multi-group confirmatory factor analysis (CFA) to understand brand inclusiveness and exclusivity across different national contexts. This analysis was critical in ensuring the comparability of our constructs across diverse cultural settings, leveraging various statistical indices to affirm measurement invariance and thus the reliability of our findings. In our study, we achieved full metric invariance across country/region samples, indicating that the way consumers perceive and respond to the factors of brand inclusivity/sustainability and brand exclusivity/trendiness is consistent across different countries/regions. This means that the factor loadings, which represent the strength and direction of the relationship between the observed variables and their underlying latent factors, are equivalent across groups. Therefore, we conclude that the factors identified have the same meaning across these groups.

However, we only attained partial scalar invariance, which suggests that most but not all item intercepts are consistent across countries/regions. Specifically, we had to relax the equality constraint for the intercept of one item (related to brand collaboration with celebrities/influencers) for the China sample. This indicates that, although the underlying concept of the factors is understood similarly across countries/regions, the actual level at which respondents from different countries/regions start to endorse the items can vary for certain items (in this case, the perception of brand exclusivity/sustainability associated with celebrity collaboration in China, which scored relatively higher in this country; i.e., consumers in China are even more willing to pay for brands that collaborate with celebrities).

Our study addresses research question 3 by exploring whether there are noticeable differences in willingness to pay (WTP) for sustainable and inclusive brands versus exclusive and on-trend brands across countries, with a particular focus on the distinction between established and emerging economies (RQ3a). We discovered significant variations: consumers in Singapore and South Korea displayed the highest WTP for inclusive and sustainable brands, which may reflect a strong cultural emphasis on ethical business practices and sustainability in these nations. On the other hand, consumers in China, South Africa, and Thailand showed a pronounced preference for exclusive brands, which are often associated with luxury and status. In contrast, Belgium, Germany, The Netherlands, and Sweden exhibited the lowest WTP for both brand types, possibly indicating a more critical or discerning consumer base that may be less influenced by brand marketing and more by product substance. Additionally, our findings revealed a consistent positive correlation between WTP for inclusive, sustainable brands and exclusive, on-trend brands across all

countries. This suggests that the motivations behind purchasing decisions for these two seemingly different brand types may not be mutually exclusive but can coexist within consumer preferences, indicating that consumers do not necessarily prioritize one set of values over the other but rather integrate both into their purchasing decisions.

### 6.2. Theoretical Contributions, Managerial Applications, Limitations, and Future Research Suggestions

In terms of theoretical contributions, this study contributes to the growing literature stream focusing on willingness to pay for sustainability/inclusivity-focused offerings (e.g., [6–10]) and, in particular, to those sub-streams that focus on GenZ customers and generational differences (e.g., [11–15]), that consider gender differences (e.g., [16–18]), and that examine country differences (e.g., [19,20]). This study also aims to add to previous reflections related to building corporate sustainability. For example, this study elaborates on specific elements of the "Honeybee" or sustainable leadership approach [74,75]. This approach builds on the sustainable Rhineland leadership practices [76] and indicates how companies should act across 23 leadership elements to implement a corporate sustainability philosophy. One of those elements is a company's perspective on stakeholders, which is included as a foundation practice in this model. Whereas a non-sustainable "Locust" approach adopts a shareholder-first perspective, a "Honeybee" or sustainable leadership approach entails a view that everyone matters [74,75,77]. This study highlights the importance of explicitly considering the customer as the stakeholder. Other research has argued that customers are an important stakeholder to consider when implementing corporate sustainability, and that this deserves more attention in academic research and business practice [78–80]. Building and managing stakeholder-based brand equity has been considered at a theoretical level as a means to manage and increase corporate sustainability [81]. We aim to contribute to this perspective by advancing customer willingness to pay for inclusive, sustainable brand offerings as an important parameter to manage when considering the "customer" stakeholder.

In terms of managerial implications, research on willingness to pay for brands that are positioned as inclusive and sustainable is of relevance. The observation of such customer willingness to pay (that we advance) should make practitioners less doubtful of whether to adopt a sustainability leadership strategy. The research evidence presented, that premium pricing margins may be expected when addressing certain customer audiences (especially those belonging to younger generations), could help Chief Sustainability Officers (CSOs) to have more leverage in the board room when advocating the adoption of sustainability-focused action plans. In other words, we provide evidence showing that brand positioning on inclusivity and sustainability can also be considered as a viable strategy from a business margin perspective. In addition, the study's insights can guide businesses and marketers in tailoring their brand strategies to align with consumer values in diverse international markets. Understanding the balance between the appeal of inclusivity/sustainability associations and exclusivity/trendiness associations in branding can help companies navigate the complex global marketplace more effectively.

Our findings offer insightful reflections into consumer preferences related to sustainable purchasing behavior across different global markets, highlighting the nuanced interplay between WTP for inclusive, sustainable brands and exclusive, on-trend brands. However, these results should be interpreted with caution due to the cross-sectional nature of the self-reported data. As already highlighted in previous research (e.g., [18,26]), such data can be influenced by social desirability bias, where respondents might answer in a way that they perceive to be more favorable or acceptable to others, rather than reflecting their true feelings or behaviors. This aspect was not controlled for in our study, potentially skewing the responses towards more socially acceptable answers.

Furthermore, we did not account for acquiescence response style (i.e., a tendency to agree with statements regardless of their content), which can vary significantly across

cultures and countries. Our findings might be affected by systematic response biases related to cultural differences in communication styles or agreement tendencies.

As the data were initially collected in the context of applied market research (and not academic research), the items were not taken from established, pre-validated scales but were rather based on marketing consultants' input. Some items might, as a consequence, be potentially problematic because they could be considered to be 'double-barreled' questions (double-barreled items in survey research are questions that combine multiple issues into a single query, requiring a single response). However, it is unlikely that this substantially hampered the validity of our conclusions, as the confirmatory factor analysis (including tests for measurement invariance) demonstrated the internal consistency of the factors.

Future research in this area should aim to control for these factors to ensure a more accurate representation of consumer attitudes. Incorporating measures to adjust for social desirability and acquiescence response style, particularly in cross-cultural settings, would provide a more robust understanding of consumer preferences. Additionally, longitudinal studies could offer deeper insights into how the observed preferences evolve over time and in response to global economic, social, and environmental changes.

Future research should also explore the underlying motivations driving WTP for inclusive, sustainable and exclusive, on-trend brands, and should examine how these factors interact with broader societal trends and individual consumer values. This approach could uncover more detailed mechanisms of consumer decision making, providing valuable information for developing more nuanced and effective marketing strategies.

**Author Contributions:** F.G.: conceptualization, methodology, investigation, resources, writing—original draft preparation, review and editing, supervision, project administration; B.W.: conceptualization, methodology, formal analysis, investigation, writing—original draft preparation, review and editing; J.V.d.B.: conceptualization, methodology, investigation, resources. All authors have read and agreed to the published version of the manuscript.

**Funding:** This research received no external funding.

**Institutional Review Board Statement:** The study was conducted in accordance with the Declaration of Helsinki, and ethical review and approval were waived for this study as it is in line with the generic ethical protocol of the Faculty of Psychology and Educational Sciences of Ghent University; https://www.ugent.be/pp/nl/onderzoek/ec/algemeen_ethisch_protocol.pdf, accessed on 8 April 2024).

**Informed Consent Statement:** Informed consent was obtained from all subjects involved in the study.

**Data Availability Statement:** The datasets presented in this article are not readily available because they are privately owned market research data. Requests to access the datasets should be directed to joeri@wearehuman8.com.

**Conflicts of Interest:** The authors declare no conflict of interest. Author Joeri Van den Bergh was employed by the company Human8. The remaining authors declare that the research was conducted in the absence of any commercial or financial relationships that could be construed as potential conflicts of interest.

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
