# Peer review of "Are Consumers Equally Willing to Pay More for Brands That Aim for Sustainability, Positive Societal Contribution, and Inclusivity as for Brands That Are Perceived as Exclusive? Generational, Gender, and Country Differences"

_sustainability, doi:10.3390/su16093879_

Round 1

Reviewer 1 Report

Comments and Suggestions for Authors

Congratulations for a great piece of work done.  Given that it is a significant piece of work, it should offer some contributions to theory/model building in this specific area of sustainability brand and the corporate sustainability in general.  The following are some ideas to improve the paper.

First, please consider adding a section to highlight knowledge gaps and contributions of the present study.  One area that it clearly contributes to is sustainability or stakeholder-based brand equity as part of the theory of corporate sustainability.  Increasingly, stakeholder-based brand equity is used as a proxy for corporate sustainability.  That means stakeholder-based brand equity model theoretically leads to improving the prospect of corporate sustainability.  The present study essentially contributes to the continuing efforts into building the "theory of corporate sustainability".  It will be a pity if the present study of a high level of external validity doesn't include this discussion since the existing theory of corporate sustainability does not include customer willingness to pay as part of the theory.

Secondly, in your discussions of the findings, please also highlight how your findings contribute to the existing relevant theories/concepts in the domain.  There are a number of concepts and theories I can think of.  For example, the Sustainable/Honeybee Leadership model includes brand and reputation as two outcomes of sustainability practices.  It doesn't touch upon customer willingness to pay, casting some doubts for the practitioners why they should adopt the sustainable leadership concept if there is no guarantee that it will increase the willingness to pay from their customers.

Please also add a discussion on how you findings offer some managerial implications for Chief Sustainability Officers since they will greatly benefit from your work.

Comments on the Quality of English Language

look fine.

Author Response

REPLY TO REVIEWER 1

_____________

Thank you for the kind words that mention the appreciation of our work. We also appreciate that you share tangible ideas on how to increase the quality of our manuscript. We took these comments to heart and used them to optimize our manuscript. Specifically, in response to the comment “to consider adding a section to highlight knowledge gaps and contributions of the present study” including possible references to “stakeholder-based brand equity” and “the Honeybee or sustainable leadership approach”, in the context of implementing “corporate sustainability” and the comment to voice “managerial implications for Chief Sustainability Officers”, we included the following additional paragraphs in our manuscript:

_____________

In terms of theoretical contributions, this study contributes to the growing literature stream focusing on willingness-to-pay for sustainability/inclusivity-focused offerings (e.g., Berger,2019; Kamboj and Matharu, 2021, Kovacs and Keresztes, 2022; Laroche, Bergeron and  Barbaro-Forleo, 2001; Wei, Ang and Jancenelle, 2018) and in particular to those sub-streams that focus on GenZ customers and generational differences (e.g., Gomes, Lopes and Nogueira, 2023; Ham, Chung, Kim, Lee and Oh, 2022; Narayanan, 2022; Squires, 2019, Yadav and Pathak, 2016), that consider gender differences (e.g., Hustvedt and Bernard, 2008; Khan, Siddiquei, Muneeb and Farhan, 2022; Shahsavar and Kube, 2020) and that examine country differences (e.g., Gregory-Smith, Manika and Demirel, 2017; Shehaw, Agag, Alamoudi, Alharthi, Brown, Labben and Abdelmoety, 2014). This study also aims to add to previous reflections related to building corporate sustainability. For example, this study elaborates on specific elements of the “Honeybee” or sustainable leadership approach (Avery and Bergsteiner, 2010; Avery and Bergsteiner, 2011). This approach builds on the sustainable Rhineland leadership practices (Avery,2005) and indicates how companies should act across 23 leadership elements to implement a corporate sustainability philosophy. One of those elements is a company’s perspective on stakeholders, which is included as a foundation practice in this model. Whereas a non-sustainable “Locust” approach adopts a shareholder-first perspective, a “Honeybee” or sustainable leadership approach entails a view that everyone matters (Avery and Bergsteiner, 2010; Avery and Bergsteiner, 2011; Kantabutra and Avery, 2013). This study highlights the importance of explicitly considering the customer-stakeholder. Other research has argued that customers are an important stakeholder to consider when implementing corporate sustainability, that deserves more attention in academic research and business practice (Chabowski et al., 2011; Goedertier et al., 2024; Simpson and Radford, 2014). Building and managing stakeholder-based brand equity has been considered at a theoretical level as a means to manage and increase corporate sustainability (Winit & Kantabutra, 2022). We aim to contribute to this perspective by advancing customer willingness-to-pay for inclusive, sustainable brand offerings as an important parameter to manage when considering the “customer” stakeholder.

In terms of managerial implications, research on willingness-to-pay for brands that are positioned as inclusive and sustainable is of relevance. The observation of such customer willingness-to-pay (that we advance), should make practitioners less doubtful whether to adopt a sustainability leadership strategy. The research evidence presented that premium pricing margins may be expected when addressing certain customer audiences (especially those belonging to younger generations), could help Chief Sustainability Officers (CSO’s) to have more leverage in the board room when advocating the adoption of sustainability-focused action plans. In other words, we provide evidence showing that brand positioning on inclusivity and sustainability can also be considered as a viable strategy from a business margin perspective. In addition, the study's insights can guide businesses and marketers in tailoring their brand strategies to align with consumer values in diverse international markets. Understanding the balance between the appeal of

inclusivity/sustainability associations and exclusivity/trendiness associations in branding can help companies navigate the complex global marketplace more effectively.

(See revised manuscript p.16, lines 582-617)

Additional references included
(apart from those highlighted on the previous page that were already present in the manuscript):

Avery, G.C. Leadership for Sustainable Futures: Achieving Success in a Competitive World. Cheltenham: Edward Elgar 2005, 263p.

Avery, G.C. & Bergsteiner, H. Honeybees & Locusts: The Business Case for Sustainable Leadership. Sydney: Allen & Unwin 2010, 288p. (republished in 2011 as Avery, G.C. & Bergsteiner, H. Sustainable Leadership: Honeybee and Locust Approaches. International Version, Routledge).

Avery, G.C. & Bergsteiner, H. Sustainable leadership: practices for enhancing business resilience and performance. Strategy & Leadership 2011, 39 (3), 5-15.

Chabowski, B. R., Mena, J. A., & Gonzalez-Padron, T. L The structure of sustainability research in marketing, 1958–2008: A basis for future research opportunities. Journal of the Academy of Marketing, Science 2011, 39(1), 55–70.

Goedertier, F., Weijters, B., Van den Bergh, J. & Schacht,O. What does sustainability mean in the minds of consumers? A multi-country panel study. Marketing Letters 2024, in press.

Kantabutra, S. & Avery, G.C. Sustainable leadership: Honeybee practices at a leading Asian industrial. Asia-Pacific Journal of Business Administration 2013, 5(1), 36-56.

Simpson, B. J., & Radford, S. K. Situational variables and sustainability in multi-attribute decision-making. European Journal of Marketing 2014, 48(5–6), 1046–1069.

Winit, W. & Kantabutra, S. Enhancing the Prospect of Corporate Sustainability via Brand Equity: A Stakeholder Model. Sustainability 2022, 14(9), 4998.

Reviewer 2 Report

Comments and Suggestions for Authors

Introduction

·       It's challenging to identify the research problem in this article. Initially, the author presents an illustration indicating disparities in the willingness to purchase a product. The researcher highlights this distinction by juxtaposing research findings with content from a Forbes article. However, what exactly constitutes the research problem in this context remains unclear.

·       Researchers utilize research questions instead of research hypotheses. However, a research hypothesis provides a more specific direction for the study. It is suggested that researchers, having reviewed prior research findings, should be capable of formulating research hypotheses instead of posing general research questions.

Research Methods

·       Where do the research indicators originate from? The researcher did not cite the source of the research indicators used in the questionnaire.

·       Researchers employed quota sampling to select respondents from each generation. However, there is a need for researchers to justify the unequal distribution of respondents across generations. Why is the number of respondents for Generation Z consistently almost double compared to Generations X and Y?

·       There was a double-barreled bias in the items used (Table 2. Items by factor).

Results and Discussion

·       The research results section is challenging to comprehend as the researcher fails to connect the analysis findings with the research questions. Similarly, this issue extends to the discussion section.

Conclusions

·       Present the research conclusions. The subtitle 'Research Reflection' was considered inappropriate, as the researcher merely stated that the study offered reflection on consumer preferences without actually conveying the reflection itself.

Author Response

REPLY TO REVIEWER 2

_____________

Thank you for taking the time to review our paper in-depth! We first address your comments formulated under the heading “introduction” in your reviewer notes. In response to the comment that signals the opportunity to include a more explicit research problem formulation, we added the text lines highlighted below in the latest version of our manuscript. We hope that this addition makes it easier for the uninformed reader to grasp the focus of our study early-on in the paper.

_____________

“Summarizing, the research need that this study addresses is the necessity of advancing additional evidence of consumer willingness-to-pay for inclusive, sustainable offerings of specific customer groups (to increasingly trigger brand positioning on inclusivity and sustainability by businesses). Specifically, we aim to contribute by examining whether “inclusive” brand positioning can trigger similar increased willingness-to-pay as “exclusive” brand positioning. And if so, for which customer groups especially.”

(See revised manuscript p.2, lines 67-73)

_____________

In response to the comment about the use of research questions or hypotheses, we want to indicate that we fully acknowledge the value of using hypotheses. In fact, we debated among co-authors on which approach to use (hypotheses vs research questions) and concluded that there are “pros” and “cons” to both approaches. While there is indeed a clear value in using hypotheses to provide a specific direction for the study, we decided in the end to adopt a more generic approach using research questions as some aspects of our study include a diverse data range (e.g., we analyze data from 20 countries). To avoid a possible overload of hypotheses (e.g., one for each country or specific region) and a subsequent complex write-up of our results section (and some other sections), we opted for a research question approach. We wanted to clarify this reasoning (and also acknowledge at the same time the value of a “hypotheses” set-up as already mentioned). It adds value to bring the point of “providing specific directions” when/where possible to our attention. We are thankful for making this point salient in our minds and will consider it in future research initiatives.
_____________

_____________

In terms of the comments under the heading “research methods”, we regret that the origin/conceptualization process of the research questions and the reason for over-sampling GenZ respondents was unclear for you when reading the first version of our manuscript. Thank you for explicitly mentioning this. We highlight below the text sections that are included in the latest manuscript (containing add-ons compared to the first manuscript version) to address these concerns.
_____________

“…The survey then proceeded with a series of questions relating to general consumption behaviors and preferences. These questions were generated starting from the insights of previous qualitative and quantitative studies (i.e., a  qualitative study based on 26 expert interviews with senior marketing executives of youth targeting brands (Van den Bergh and Pallini, 2017), a quantitative study involving 10,000 respondents originating from 8 European countries (Van den Bergh, 2018),) and an interpandemic qualitative study involving 16- to 19-year-olds (N=200) originating from 8 European countries during a 3 week online community (Van den Bergh et al., 2020)).”

(See revised manuscript p.10, lines 384-390)

Additional references included:

Van den Bergh, J. & Pallini, K. Fragile. Is Next Gen marketing more chemistry than science? InSites Consulting 2017. Available online: https://www.insites-consulting.com/bookzines/fragile/ (accessed 26 April 2024).

Van den Bergh, J. How brands can effectively engage young consumers. WARC 2018. Available online: https://www.warc.com/content/paywall/article/bestprac/how-brands-can-effectively-engage-young-consumers/en-gb/122479? (accessed 26 April 2024).

Van den Bergh, J., Quaschning, S., Zhuk, Y. & Goderich, D. The impact of Covid-19 on the world of teens. Research report commissioned by The Coca-Cola Company WEBU & CEE 2020. Article about this report available online: https://www.linkedin.com/pulse/gen-z-unmasked-how-pandemic-affecting-teens-worldwide-van-den-bergh/ (accessed 25 April 2024).

_____________
Thank you for also raising the potential issue of a double-barrelled bias in some items used. In response to this comment, we included an explicit mention in the limitations section related to this observation. As such, we hope to draw attention to the fact that taking a nuanced, critical perspective related to this element is
valuable. See the text fragment highlighted below.……………………………………………………………………
_____________

“As the data was initially collected in the context of applied market research (and not academic research), the items were not taken from established, pre-validated scales, but were based on marketing consultants’ input. Some items might as a consequence be potentially problematic because they could be considered to be ‘double-barreled’ questions (double-barreled items in survey research are questions that combine multiple issues into a single query, requiring a single response). However, it is unlikely that this substantially hampered the validity of our conclusions, as the confirmatory factor analysis (including tests for measurement invariance) demonstrated the internal consistency of the factors.

(See revised manuscript p.17, lines 630-637)

_____________

Finally, in response to the comment under the heading “results and discussion”, we tried to connect the research questions better (more explicitly) with the observations. Specifically, we now start each of the paragraphs that are linked to one of our research questions by explicitly referring to it:
_____________

“Our study directly addresses the research question concerning whether younger generations, particularly GenZ, show a greater willingness to pay (WTP) for brands that emphasize sustainability, positive societal impact, and inclusivity (RQ1).” … “Our research specifically addresses the second research question by examining whether, […]”

“As to research question 3, […]” and “Our study addresses Research Question 3 by […]”

(See revised manuscript p.15, lines 514-516,529-530,541,563)

Reviewer 3 Report

Comments and Suggestions for Authors

The presented article offers a remarkable contribution to the literature. The size of the survey is remarkable, key countries with different consumption behaviours are analysed, giving the possibility to analyse the results from different points of view. It is a pity that some iconic countries in purchasing styles such as Italy are missing, also due to the fact that the concept of Made in Italy is mentioned. 

In this article that I am including here, you can take some results on the difference between perception and attitude that can certainly enrich the literature, especially in the discussion part for the country Argentina. 

Damico, A. B., Vecchio, Y., Masi, M., & Di Pasquale, J. (2023). Perceptions and Attitudes of Argentine Zoomers towards Sustainable Food Production. Foods, 12(5), 1019.

From line 385 to line 404 we talk about methodology, move this part, and explain SEM better, also to better understand the relationships that have been determined. It is a little difficult to understand where the explanation of the method ends and where the results part begins, reorganise better.

Author Response

REPLY TO REVIEWER 3

_____________
Thank you for reviewing our manuscript. We are very appreciative of your praise for some aspects of our study. In response to your comment about the omission of Italy as a country in which respondents were recruited and analyzed, we want to signal that we would also have loved to have included this region. However, the selection of countries/regions was restricted/determined by the research agency that we collaborated with. As there is a substantial cost involved when adding countries/regions, the research agency was willing to “sponsor” the data collection exclusively for countries/regions that were of direct commercial value to them (given ongoing activities/client bases there). In the latest version of the manuscript, we added a text line referring to this “restriction” (see the last sentence add to the paragraph depicted below).

_____________

“For the current study we analyzed secondary data from a large-scale cross-national online survey conducted among the online panel of a globally operating market research and consulting agency using quota sampling for country, gender, and age cohort (oversampling GenZ as this generation was a core focus of the survey study). The company selected the countries based on commercial relevance.”

(See revised manuscript p.8, lines 375-376)

_____________
Thank you for sharing a specific, recent article in your reviewer reply that focuses on “Zoomer” perceptions and attitudes related to sustainably produced products. We included it as an additional reference in our paper! See also the text line highlighted below that was added in the latest version of the manuscript.
_____________

“Damico et al. (2023) survey GenZ “Zoomer” consumers in Argentina and observe that they express a high concern for the health of the planet and for unsustainable production methods.”

(See revised manuscript p.4, lines 172-174)

Additional reference included:

Damico, A. B., Vecchio, Y., Masi, M., & Di Pasquale, J. Perceptions and Attitudes of Argentine Zoomers towards Sustainable Food Production. Foods 2023, 12(5), 1019.

____________
Finally, in response to the comment related to the explanation and organization of the methods and results section, we moved the CFA table (Table 2) to the Results section and listed the items in the text (to avoid cross-referencing between methods and results). In addition, to enhance readability, we rewrote the paragraph on the SEM as follows: ………………………………………………………………………………………………………..
_____________

“As the next step, we develop a multi-group Structural Equation Model, treating "brand inclusiveness" and "brand exclusivity" as outcomes influenced by demographic variables. Specifically, we use dummy variables for gender (1 representing female, 0 representing male) and age groups (dummies for Generation Y, Generation X, and Baby Boomers). To prevent the model from becoming too complex and to ensure it remains straightforward, we check if the influence of these demographic variables on the brand factors is consistent across different countries. This process, known as testing for invariance, helps us ensure that the model is not overly tailored to specific data sets. As indicated in Table 3, and based on the Bayesian Information Criterion (BIC), the relationships between demographics and brand factors do not vary by country, suggesting a stable model across various national contexts.”

(See revised manuscript p.12, lines 446-456)